# Real-World Use of Dalbavancin in Diabetic Foot Osteomyelitis: PEDDAL Study

**DOI:** 10.3390/jcm14196705

**Published:** 2025-09-23

**Authors:** Leonor Moreno Núñez, Ana Rueda Benito, Elena Bereciartua, Laura Morata, Rosa Escudero-Sánchez, Julia Sellares Nadal, Joan Gómez-Junyent, Alejandro Salinas Botrán, Ana María Arnáiz García, María Dolores del Toro López, Diana Ruiz-Cabrera, José Miguel Ramos Andrino, Fatma Alidrous, Marina Torío-Salvador, Miguel Ángel Verdejo, María Velasco Arribas

**Affiliations:** 1Infectious Diseases Department, Hospital Universitario Fundación Alcorcón, Universidad Rey Juan Carlos, 28922 Madrid, Spain; 2Microbiology Department, Sligo University Hospital, F91H684 Sligo, Ireland; 3Infectious Disease Department, Cruces University Hospital, Biobizkaia Research Institute, 48903 Baracaldo, Spain; 4Infectious Diseases Department, Hospital Clinic-IDIBAPS, Universitat de Barcelona, 08036 Barcelona, Spain; 5Infectious Disease Department, Hospital Ramón y Cajal (Madrid), IRYCIS (Instituto Ramón y Cajal de Investigación Sanitaria), CIBERINFEC (Centro de Investigación Biomédica en Red Enfermedades Infecciosas), 28034 Madrid, Spain; 6Infectious Disease Department, Hospital Universitari Vall d’Hebron, Vall d’Hebron Barcelona Hospital Campus, 08035 Barcelona, Spain; 7Department of Infectious Diseases, Hospital del Mar, Institut Hospital del Mar d’Investigacions Mèdiques (IMIM), Universitat Pompeu Fabra, 08003 Barcelona, Spain; 8Internal Medicine Department, Hospital Clínico San Carlos, Universidad Complutense de Madrid, 28040 Madrid, Spain; 9Internal Medicine Department, Gerencia de Atención Especializada, Áreas III y IV: Hospital Sierrallana y Tres Mares, Servicio Cántabro de Salud, 39300 Torrelavega, Spain; 10Unidad Clínica de Enfermedades Infecciosas y Microbiología, Hospital Universitario Virgen Macarena, Universidad de Sevilla, Centro de Investigación Biomédica en Red en Enfermedades Infecciosas (CIBERINFEC), Instituto de Salud Carlos III, 41009 Madrid, Spain; 11Research Unit, Hospital Universitario Fundación Alcorcón, GEIO-SEIMC, 28922 Madrid, Spain

**Keywords:** diabetic foot infection, osteomyelitis, dalbavancin, effectiveness, multidrug-resistant infections

## Abstract

**Background:** Data on antibiotic treatment for diabetic foot osteomyelitis are limited. This study aims to describe the real-world effectiveness and safety of dalbavancin in treating deep diabetic foot infections. **Methods:** A retrospective, observational, multicenter study was conducted in nine Spanish hospitals and one Irish hospital. Patients with diabetic foot osteomyelitis treated with dalbavancin were included. Data on demographics, clinical characteristics, microbiology, antibiotic regimens, adverse events, and clinical outcomes were analyzed. **Results:** Among 136 patients, 76% were male, with a mean age of 69 ± 12 years. Renal insufficiency was observed in 32%, and 6% required renal replacement therapy. Based on the McCabe scale, 70% of patients had a rapidly or ultimately fatal disease. Polypharmacy was noted in 83%, and 60% of infections were moderate. Dalbavancin was primarily used as second-line therapy (92%). The cure rate was 80.9% (95% CI: 73.5–86.6%), achieved after a median of two doses. Patients receiving dalbavancin as first-line therapy had a cure rate of 86%, comparable to 80% in second-line therapy, with no significant differences. Surgical interventions were required in 72% of cases, with minor amputations performed in 40% of patients. Polymicrobial infections were common (55%), and methicillin-resistant Staphylococcus aureus was identified in 26% of cases. Adverse events occurred in 5% of patients. Chronic kidney disease was the sole independent risk factor for therapeutic failure (IRR 0.80, 95% CI: 0.64–1.00, *p* = 0.045). **Conclusions**: Dalbavancin is effective and safe for treating diabetic foot osteomyelitis, including in complex patients with resistant microorganisms.

## 1. Introduction

In Europe, approximately 1 in 10 adults aged between 20 and 80 years is diabetic [1], and the prevalence of type 2 diabetes has been increasing over the past three decades. One of the long-term complications of diabetes is diabetic foot disease, which is the most common cause of hospital admissions among these patients [2]. Osteomyelitis is the most frequent infection in diabetic foot ulcers, occurring in >20% of moderate infections and 50–60% of severe infections, and it is associated with high amputation rates [3]. Furthermore, diabetic foot osteomyelitis is considered a complex and challenging infection to treat, with a high relapse rate [4], requiring multidisciplinary teams for proper management. In our environment [5], at least one species of Gram-positive bacteria is present in 91% of bone cultures, with *S. aureus*, coagulase-negative staphylococci, and *Corynebacterium* spp. being the most prevalent microorganisms. Additionally, there is a high prevalence of methicillin resistance in *S. aureus* (15–30%) [6], which is considered a risk factor for hospitalization [7], and, along with other multidrug-resistant microorganisms, influences mortality and amputation risk [8].

Antibiotic treatment is the cornerstone of diabetic foot infections. However, the rise in antibiotic resistance may lead to increased therapeutic failure. In this context, the development of new antibiotics [9] can be a valuable tool for treating deep diabetic foot infections. Dalbavancin is a glycopeptide used to treat Gram-positive microorganism infections with unique pharmacokinetic properties. Clinical trials have demonstrated its efficacy and safety in treating skin and soft tissue infections [10], as well as adult osteomyelitis [11,12]. Additionally, dalbavancin has shown high efficacy in diabetic patients with skin and soft tissue infections and other hard-to-treat infections with a favorable safety profile [13]. However, studies on the use of dalbavancin in deep diabetic foot infections are rare, with only one study from our group involving 23 patients, reporting an 87% cure rate [14], and a single case reported from Greece [15].

The aim of this study is to describe the real-world effectiveness and safety of dalbavancin in deep diabetic foot infections.

## 2. Materials and Methods

This was a multicenter, multinational, retrospective, observational study of patients with deep diabetic foot infections (osteomyelitis) treated with dalbavancin from its approval until April 2023. Patients were included from nine Spanish hospitals and one in Ireland who received dalbavancin for diabetic foot infection as a first-line treatment or as rescue therapy, with follow-up until the end of the episode in participating hospitals, and the dalbavancin administration provided occurred before the study’s approval by the Ethics Committee (Institutional Review Board).

Patients were identified through dalbavancin prescription or usage records from hospital pharmacies or patient registries. Eligibility required infection signs based on the Infectious Diseases Society of America (IDSA) criteria [16], targeting Gram-positive microorganisms treated with dalbavancin as first-line therapy or due to treatment failure, toxicity, or drug interactions with prior antibiotics (e.g., cotrimoxazole, quinolones, oxazolidinones, and tetracyclines). Treatment decisions and dosing were made by infectious disease specialists routinely involved in diabetic foot care or members of multidisciplinary Diabetic Foot Units.

Osteomyelitis was diagnosed using PTB positivity combined with radiography, with or without bone culture according to IDSA guidelines [16,17,18]. Peripheral arterial disease was diagnosed based on clinical examination and/or imaging studies performed in routine care (e.g., Doppler ultrasound), as per the local protocols. Neuropathy was inferred from clinical history and physical exam findings, such as reduced sensitivity or neuropathic pain. The classification of infection severity followed IDSA guidelines (moderate vs. severe infections). We used standard clinical definitions without employing a specific wound grading system. Deep infection was operationally defined as osteomyelitis, not including soft tissue infections. Diagnosis was established per IDSA guidelines, combining probe-to-bone testing, plain radiography, and bone culture when available. MRI was not required.

Cure was defined as a wound remaining healed for 90 days after primary closure or, for patients requiring amputation, a negative intraoperative culture after at least one dose of dalbavancin. Therapeutic failure was defined by persistent infection signs per IDSA criteria [16], persistent positive probe-to-bone (PTB) testing, or non-healing wounds.

Demographic variables, comorbidities based on the Charlson index [19], and the McCabe index (prognostic classification as rapidly fatal, ultimately fatal, or non-fatal) [20], and regular medications (≥5 drugs to define polypharmacy [21]) were recorded to predict adverse drug interactions. Infection-related variables included severity (moderate, severe per IDSA [16]), type (ischemic, neuropathic, mixed), osteomyelitis diagnostic method, and microbiological findings (bone culture, blood culture, wound exudate). Difficult-to-treat bacteria included methicillin-resistant *S. aureus*, *Corynebacterium* spp., and *Enterococcus faecium*. Surgical interventions, need for revascularization, and negative pressure therapy were also recorded. Previous antibiotic therapy for the lesion later treated with dalbavancin, its indications, dosages, and adverse events (classified per international criteria [22]) were documented.

Statistical analysis: To describe the variables we used the mean and standard deviation (SD) or percentage as appropriate, and comparisons between groups were made with the Student’s *t*-test, Mann–Whitney, or Chi square as appropriate. We used modified Poisson regression models to assess the risk factors associated with cure rates, with each of these outcomes as the dependent variable, incorporating clinically relevant or statistically significant variables (*p* < 0.05) identified in the exploratory univariate analysis. Missing data were minimal and related primarily to comorbidity scores. These cases were excluded from specific analyses. The software Stata v17 was used for the analysis.

The study adhered to good clinical practice and the Declaration of Helsinki and was approved by the Institutional Review Board (Ethics Committee) at Fundación Alcorcón University Hospital (approval code: GNJ-DAL-2020-01).

## 3. Results

A total of 136 patients were included, of whom 76% (103/136) were male, with a mean age of 69 years ± 12 SD. Among them, 32% (44/136) had renal insufficiency, and 6% were receiving renal replacement therapy. Fatal or rapidly fatal disease was observed in 70% (81/116) according to the McCabe classification, with a mean Charlson index of 4 ± 2 SD. McCabe data were unavailable in 20 cases due to documentation gaps. Polypharmacy was noted in 83% (113/136) of patients. The infection was classified as moderate in 60% (81/136) of cases, with ischemic diabetic foot being the most prevalent type (42%, 57/136). Polymicrobial infections were identified in 55% (75/136), with the most frequently isolated dalbavancin-sensitive microorganism being methicillin-sensitive *Staphylococcus aureus* (29%, 51/176 samples), followed by methicillin-resistant *Staphylococcus aureus* (26%, 46/176) and *Corynebacterium* spp. (18%, 32/176). The remaining clinical, microbiological, and diagnostic characteristics are shown in Table 1. Only dalbavancin-sensitive microorganisms are indicated.

In 92% (123/134) of patients, dalbavancin was used as second-line therapy, with the majority of cases (78%, 106/116) involving targeted use (microbiologically guided treatment). The most common reasons for dalbavancin use were prior antibiotic toxicity (27%, 31/116), poor progression of the infection (24%, 28/116), expediting hospital discharge (22%, 25/116), avoiding hospital admission (12%, 14/116), and avoiding drug interactions (9%, 10/116). The antibiotics most frequently used prior to dalbavancin prescription were linezolid (20%, 50/246), quinolones (17%, 41/246), and intravenous beta-lactams (15%, 38/246). Before starting dalbavancin, 35% (46/130) of patients had received two types of antibiotics, and 18% (24/130) had received three types, with a median duration of prior antibiotic therapy of 28 days (IQR 11–50). The most commonly used dalbavancin regimen was 1500 mg every two weeks for two doses (45%, 61/136), followed by a single 1500 mg dose (20%, 27/136). In 32% (43/136) of cases, dalbavancin was prescribed alongside another antibiotic, most commonly a quinolone (59%, 25/43).

Adverse effects were reported in 5% (7/136) of patients, most of which were mild. Three patients experienced gastrointestinal symptoms (diarrhea and nausea), leading to dalbavancin discontinuation in three cases. Two patients developed a skin rash, and two experienced hyperglycemia during dalbavancin infusion. Two patients experienced chills during the drug administration. Lastly, one patient died between the first and second doses of dalbavancin due to SARS-CoV-2 infection. Adverse effects are detailed in Table 2.

In addition to dalbavancin antibiotic treatment, 77.2% (105/136) of patients required surgical intervention. Minor amputation with or without revascularization was performed in 40% (55/136) of cases, and only revascularization in 14% (19/136). Revascularization was deemed unfeasible in 11 patients who required it. The surgical procedures performed are detailed in Table 3.

At 90 days post-dalbavancin treatment, 80.9% (110/136) of patients achieved a cure (95% CI: 73.5–86.6%). In the univariate analysis, cure rates were lower among patients with a higher Charlson index (5.3 vs. 4.2, *p* = 0.02) and those with chronic kidney disease (>60 severe vs normal: 50.0% vs. 83.3%, *p* = 0.047). Multivariate analysis, which included chronic kidney disease, hepatic disfunction, chronic heart failure, sex, age, and infection with difficult-to-treat bacteria, identified the chronic kidney disease as the only statistically significant independent risk factor for therapeutic failure with dalbavancin (IRR 0.80, CI 95% 0.64–1.00 *p* = 0.045), as shown in Table 4. We did not find association between cure rates and Charlson index in the multivariate analysis.

## 4. Discussion

Dalbavancin is an effective and safe antibiotic for the treatment of osteomyelitis in adults [12]. Our study represents one of the largest multicenter real-world studies on the use of dalbavancin for diabetic foot osteomyelitis. In our study, the cure rate of dalbavancin combined with standard surgical management was high, at 80.9% (95% CI: 73.5–86.6%), consistent with results published by other authors using similar cure criteria to ours [23].

Diabetic foot osteomyelitis is a complex infection with frequent persistence and relapses [4]. This complexity is attributed to several factors, including immunological deterioration and reduced inflammatory response, which primarily occur in necrotic bone. Additionally, the activity and number of leukocytes decrease when microorganisms adhere to biofilms. Moreover, diabetes mellitus itself complicates the resolution of infections [24], as hyperglycemia promotes immune dysfunction, deterioration of the antioxidant system, and impairment of humoral immunity. These factors likely contribute to the slightly lower cure rate observed in our study compared to other studies with a limited proportion of diabetic foot infections [13,25], where only 14–33% of patients with osteomyelitis treated with dalbavancin had diabetes [11,12,26].

In our cohort, a quarter of patients (24%) started dalbavancin treatment following the failure of prior therapeutic regimens. These patients had received 2–3 different antibiotics for a median of 28 days before starting dalbavancin, indicating its use in chronic, complex, and difficult-to-treat infections where prior antibiotics had failed. This suggests that earlier use of dalbavancin could lead to better clinical outcomes; however, the data are observational and not derived from a randomized clinical trial. Additionally, early intervention might reduce drug interactions, as 83% of patients in our cohort were on polypharmacy [14], and mitigate antibiotic toxicity, which was the reason for dalbavancin use in 26% of cases in our study—a proportion similar to that reported in other studies [26].

In most published studies [10,13,26,27,28], the primary indication for dalbavancin was to simplify antibiotic regimens, often to expedite hospital discharge, reduce hospital stays, and lower costs related to hospitalization. In other studies [29], dalbavancin was used to treat *S. aureus* bacteremia in patients with barriers to accessing adequate healthcare, ensuring high adherence to treatment. In our cohort, most patients prescribed dalbavancin were managed in outpatient settings. Thus, its primary utility was to optimize prior antibiotic treatment (either due to failure or toxicity) and prevent drug interactions, ensuring 100% adherence in patients with high comorbidities and polypharmacy.

Not all Gram-positive microorganisms implicated in bone infections exhibit the same sensitivity patterns. Methicillin-resistant *S. aureus*, *Corynebacterium* spp., and *Enterococcus faecium* are resistant to various antibiotic classes and are therefore challenging to treat. Furthermore, infections caused by these organisms have been associated with poorer clinical outcomes [7,8]. Published experience with dalbavancin for bone infections shows the variable prevalence of these microorganisms. For instance, in Almangour’s study [11], the proportion of methicillin-resistant *S. aureus* was high (48%), but there were no patients with infections caused by *Enterococcus faecium* or *Corynebacterium* spp. In other studies, the percentages of these bacteria are very low [12]. In our cohort, two of the three most frequently isolated microorganisms were methicillin-resistant *S. aureus* and *Corynebacterium* spp., constituting 44% of Gram-positive organisms. Additionally, infections caused by *Enterococcus faecium* were also represented. The cure rate for dalbavancin was similar for these difficult-to-treat microorganisms compared to others. Moreover, in multivariate analysis, infection with these microorganisms was not an independent risk factor for treatment failure. The lack of association between resistant pathogens and failure may be limited by power. This validates the efficacy of dalbavancin in diabetic foot osteomyelitis, even for infections caused by challenging microorganisms.

As observed in other multicenter and multinational studies [13], the dalbavancin dosing regimen in our study was highly heterogeneous. However, nearly half of the cohort received a regimen of 1500 mg every two weeks for two doses. This regimen is supported by pharmacokinetic studies demonstrating high bone concentrations 14 days after administration [30]. Dose adjustments were not made for CKD, given dalbavancin’s pharmacokinetics. Experimental data also suggest that the efficacy of long half-life drugs, such as dalbavancin, increases when higher doses are administered earlier in the treatment course. Based on the efficacy demonstrated in this study and its ease of use, we believe this regimen could become a very good choice for osteoarticular infections, in contrast to regimens used for other infections [26,27]. We propose an early sequencing strategy for dalbavancin to minimize drug interactions and toxicity, administering 1500 mg every two weeks for two doses until better pharmacokinetic data become available.

As shown in other studies [13], dalbavancin demonstrated an excellent safety profile, with only 5% of patients experiencing adverse effects, most of which were mild.

Our study has limitations inherent to retrospective studies and the heterogeneity of infection management due to its multicenter nature. However, it successfully recruited a large number of complex patients with high comorbidities and deep diabetic foot infections, an area where scientific evidence is scarce. The definition of cure is set at 90 days, which represents a relatively short follow-up period for osteoarticular infections. Relapse rates beyond 90 days were not systematically assessed. Additionally, data on the timing of surgical procedures in relation to dalbavancin treatment and final cure, and the correlation between the efficacy of dalbavancin and the site of ulcers are unavailable. Future research should focus on assessing long-term outcomes of dalbavancin treatment in osteomyelitis.

## 5. Conclusions

Dalbavancin is an effective and safe antibiotic for the treatment of diabetic foot osteomyelitis. It achieves high cure rates in complex, polypharmacy-treated patients with refractory infections involving highly resistant microorganisms, maintaining a favorable safety profile. Chronic kidney disease was identified as the sole independent risk factor for therapeutic failure. The early use of dalbavancin as a first-line drug in selected cases could mitigate toxicity, prevent drug interactions, and ensure optimal adherence to antibiotic therapy.

## Figures and Tables

**Table 1 jcm-14-06705-t001:** Demographic, clinical, and microbiological characteristics of the cohort.

Variable	Number (%) * (n = 136)
Demographic Data	
▪ Age (mean ± SD)	69 ± 12
▪ Gender, male	103 (76%)
Comorbidities	
▪ Hypertension	105 (76%)
▪ Dyslipidemia	98 (72%)
▪ Active smoking	67 (49%)
▪ Chronic kidney disease (glomerular filtration rate < 60 mL/min)	44 (32%)
▪ Heart failure	36 (26%)
▪ Arrhythmia	32 (24%)
▪ Ischemic heart disease (myocardial infarction)	31 (23%)
▪ Stroke	18 (13%)
▪ Chronic obstructive pulmonary disease (COPD)	16 (12%)
▪ Hemodialysis	8 (6%)
▪ Moderate-severe liver disease	5 (4%)
▪ Dementia	3 (2%)
Charlson Index (median, IQR)	4 (3–6)
McCabe Index (n = 116)	
▪ Rapidly fatal disease	7 (6%)
▪ Ultimately fatal disease	74 (64%)
▪ Non-fatal disease	35 (30%)
Polypharmacy (>5 drugs)	113 (83%)
Infection Severity **	
▪ Moderate	81 (60%)
▪ Severe	54 (40%)
Type of Diabetic Foot (n = 116)	
▪ Ischemic	57 (49%)
▪ Neuroischemic	41 (35%)
▪ Neuropathic	18 (15%)
Osteomyelitis Diagnosis	
▪ Positive PTB	95 (70%)
▪ Bone culture	111 (82%)
▪ Simple radiography	76 (56%)
Type of Sample for Culture (n = 116)	
▪ Exudate	36 (29%)
▪ Blood cultures	18 (14%)
▪ Bone	72 (57%)
Polymicrobial Infection	75 (55%)
Microorganisms (n = 176) ***	
▪ *Methicillin-sensitive Staphylococcus aureus*	51 (29%)
▪ *Methicillin-resistant Staphylococcus aureus*	46 (26%)
▪ *Corynebacterium* spp.	32 (18%)
▪ *Coagulase-negative staphylococci*	15 (9%)
▪ *Streptococcus* spp.	15 (9%)
▪ *Enterococcus faecalis*	14 (8%)
▪ *Enterococcus faecium*	2 (1%)

* Data are expressed as n (%) unless otherwise indicated. ** According to IDSA guideline classification. *** Only dalbavancin-sensitive microorganisms are indicated. PTB: probe to bone. Some patients had polymicrobial infections, resulting in more isolates than patients.

**Table 2 jcm-14-06705-t002:** Characteristics of dalbavancin prescription.

Variable	Number (%) * (n = 136)
Type of Treatment (n = 116)	
▪ Empirical	10 (9%)
▪ Targeted	106 (91%)
First-Line Treatment **	11 (8%)
Rescue Treatment	123 (92%)
Reason for Indication (n = 116)	
▪ Toxicity from previous antibiotic therapy	31 (27%)
▪ Treatment failure	28 (24%)
▪ To expedite hospital discharge	25 (22%)
▪ To avoid hospital admission	14 (12%)
▪ To avoid drug interactions	10 (9%)
▪ Other	8 (7%)
Previous Antibiotics (n = 246)	
▪ Linezolid	50 (20%)
▪ Quinolone	41 (17%)
▪ Intravenous beta-lactam	38 (15%)
▪ Vancomycin or daptomycin	30 (12%)
▪ Oral beta-lactam	27 (11%)
▪ Clindamycin	24 (10%)
▪ Cotrimoxazole	23 (9%)
▪ Tetracyclines	7 (3%)
▪ Tedizolid	4 (2%)
▪ Rifampicin	2 (1%)
Number of Previous Antibiotics	
▪ One	37 (27%)
▪ Two	46 (34%)
▪ Three	24 (18%)
▪ Four	7 (5%)
▪ Five	4 (3%)
▪ Six	4 (3%)
▪ Nine	1 (0.7%)
Days of Previous Antibiotic Therapy (median, IQR)	28 (11–50)
Dosages Used	
▪ 1500 mg every two weeks (two doses)	61 (45 %)
▪ 1500 mg single dose	27 (20%)
▪ 1000 mg followed by 500 mg weekly for 5 weeks	15 (11%)
▪ 1000 mg followed by 500 mg	10 (7%)
▪ 1000 mg single dose	9 (6%)
▪ 1000 mg followed by 500 mg weekly for 3 weeks	6 (4%)
▪ 1000 mg followed by 500 mg for 9 weeks	3 (2%)
▪ 1500 mg followed by 500 mg weekly for 2 weeks	1 (1%)
▪ 1000 mg followed by 500 mg for 8 weeks	1 (1%)
▪ 500 mg weekly for 3 weeks	1 (1%)
▪ 750 mg weekly for 6 weeks	1 (1%)
▪ 1500 mg plus 500 mg weekly	1 (1%)
Adverse Effects (n = 7; 5%)	
▪ Diarrhea	1 (0.6%)
▪ Nausea	2 (1.5%)
▪ Skin rash	2 (1.5%)
▪ Hyperglycemia	2 (1.5%)
Administration Incidents (n = 2; 1.5%)	
▪ Shivering at each administration	2 (1.5%)
Reasons for Discontinuation of Dalbavancin	
▪ Cure	110 (81%)
▪ Poor progression	23 (16%)
▪ Toxicity	3 (2.4%)
▪ Supply shortage	1 (0.6%)

* Data are expressed as n (%) unless otherwise indicated. ** Data from 134 patients.

**Table 3 jcm-14-06705-t003:** Surgical interventions associated with dalbavancin treatment.

Variable	Number (%) * (n = 136)
Type of Surgical Procedure	
▪ Minor amputation	31 (23%)
▪ Minor amputation and revascularization	24 (18%)
▪ Debridement	25 (18%)
▪ Only Revascularization	19 (14%)
▪ Major amputation	6 (4%)
Type of Revascularization	
▪ Angioplasty	26 (19%)
▪ Bypass	10 (7%)
▪ Multiple types of revascularization	7 (5%)
Negative Pressure Therapy	31 (23%)
Number of Surgical Interventions	
▪ One	71 (52%)
▪ Two	20 (15%)
▪ Three	2 (1.5 %)
▪ Four	2 (1.5%)

* Data are expressed as n (%) unless otherwise indicated. Percentages are calculated based on the total number of patients, not the total number of surgeries, as not all patients underwent surgical intervention.

**Table 4 jcm-14-06705-t004:** Factors associated with cure. Univariate and multivariate analysis.

	Univariate Analysis			Multivariate Analysis **		
Variable	RR	CI 95%	*p*	RR	CI 95%	*p*
Age	1	0.99–1.01	0.956	1	0.99–1.01	0.958
Sex (women)	1.07	0.89–1.27	0.475	1.06	0.87–1.28	0.568
Charlson index	0.95	0.91–1	0.046			
Heart failure	0.86	0.69–1.07	0.18	0.90	0.72–1.13	0.366
Moderate-severe liver disease	0.49	0.16–1.43	0.191	0.51	0.18–1.44	0.207
Moderate-severe kidney disease	0.78	0.62–0.98	0.036	0.80	0.64–1.00	0.045
Difficult to treat bacteria *	0.90	0.75–1.02	0.87	0.96	0.81–1.12	0.581

* Difficult-to-treat bacteria included methicillin-resistant *S. aureus*, *Corynebacterium* spp., and *Enterococcus faecium*. RR: relative risk; CI 95% confidence interval. ** Modified Poisson regression models.

## Data Availability

Data supporting reported results can be found at Research Unit, Hospital Universitario Fundación Alcorcón, Madrid, at https://www.comunidad.madrid/hospital/fundacionalcorcon/profesionales/investigacion (accessed on 20 April 2025).

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
