# Peer review of "Real-World Use of Dalbavancin in Diabetic Foot Osteomyelitis: PEDDAL Study"

_jcm, 2025, doi:10.3390/jcm14196705_

Round 1
Reviewer 1 Report
Comments and Suggestions for Authors
Title: Real-World Use of Dalbavancin in Diabetic Foot Osteomyelitis: PEDDAL Study
The research topic is certainly relevant and general interest to the readers of the journal.
I found that there are some points to clarify. I explain my concerns in more detail below
Major comments:
-The main limitation of the study is the small sample size.
-Furthermore, the authors should describe whether there is a correlation between the efficacy of Dalbavancin and the site of ulcers.
Minor comments:
-The criteria for establishing the diagnosis of peripheral arterial disease and neuropathy are not described.
-The manuscript should describe which classification system used for DFO ulcer wound grading. Providing a brief description of this system will enhance the reader’s understanding of how the wounds were categorized and insert in table 1.
-Authors should describe site of ulcers and insert in table 1.
Author Response
We would like to express our sincere gratitude to the reviewers for their thorough and constructive evaluation of our manuscript. We greatly appreciate the time and effort invested in the review process, which has significantly contributed to improving the quality and clarity of the work.
In response, we have carefully addressed each of the reviewers' comments and suggestions. Below, we provide a point-by-point reply, indicating the changes made in the manuscript and offering clarifications where appropriate. All modifications have been incorporated into the revised version of the manuscript, with corresponding changes highlighted for clarity.
We hope that our responses and the revisions made adequately address the concerns raised and meet the expectations of both the reviewers and the editorial team.
Reviewer 1
-The main limitation of the study is the small sample size.
We acknowledge that the sample size (136 patients) may be considered modest; however, this cohort represents one of the largest real-world series of patients treated with dalbavancin for diabetic foot osteomyelitis (DFO) to date. The multicenter nature and clinical complexity of the cases, many of which involved resistant pathogens and polypharmacy, increase the relevance and external validity of the findings.
-Furthermore, the authors should describe whether there is a correlation between the efficacy of Dalbavancin and the site of ulcers.
We thank the reviewer for this important suggestion. While ulcer site was not collected systematically in all participating centers, we recognize its potential impact on prognosis and response to therapy. We have added this limitation to the discussion and will include ulcer location in future prospective studies.
Reviewer 2 Report
Comments and Suggestions for Authors
Introduction
Line 52: What is the specific geographical setting referred to in “our setting”?
Line 76: How does this study build on your previous 23-patient report?
Materials and Methods
- How was “deep diabetic foot infection” operationally defined—was it strictly limited to osteomyelitis?
- Diagnostic Criteria for DFO:
– How was the diagnosis of diabetic foot osteomyelitis established across centers?
– Was imaging (MRI), bone biopsy, or probe-to-bone testing required? - Retrospective Bias Management:
– Given the retrospective design, how did the authors handle missing data, especially regarding outcomes and adverse events? - Were the IDSA diagnostic criteria uniformly applied across all ten hospitals?
- How was follow-up standardized—was healing assessed by the same personnel or teams across sites?
- Can the authors clarify if missing data were present and how they were handled statistically?
Results
- Why is the McCabe classification only reported for 116 out of 136 patients? Were data missing for 20 patients?
- Were the types of diabetic foot infections (ischemic, neuropathic, mixed) mutually exclusive or overlapping?
- Why is the total number of microbiological isolates 176, while only 136 patients were included? Were multiple isolates per patient allowed?
- How were polymicrobial infections treated? Was dalbavancin used alone or in combination with other antibiotics?
- Is there a subgroup analysis of patients with renal impairment or fatal disease classifications, given their high prevalence?
- How did the authors define “targeted use” of dalbavancin in 78% of cases—was it microbiologically guided?
- Overlap of Indications: Can patients have more than one reason for switching to dalbavancin (e.g., toxicity + discharge)? If so, how was primary indication determined?
- Combination Therapy Rationale: In the 32% of patients receiving dalbavancin with another antibiotic, was this combination based on polymicrobial infection, or empirical broadening?
- Dosing Strategy: What factors guided the selection of single vs. dual-dose dalbavancin regimens? Were cure rates stratified by regimen?
- Adverse Events Detail: Were the adverse events attributed to dalbavancin assessed by causality tools (e.g., Naranjo scale), or based solely on clinician judgment?
- Amputation & Revascularization Impact: Were cure rates stratified by the type of surgical intervention (e.g., amputation vs revascularization vs neither)?
- Multivariate Analysis: Why was Charlson index not retained as a significant variable in multivariate analysis despite significance in univariate? Was there collinearity with CKD?
Discussion
- Dose Justification: While 1,500 mg every two weeks ×2 is proposed as optimal, were outcomes compared between dosing regimens in this study? If so, did this regimen actually show superior results?
- Microbiological Risk Factors: The claim that resistant organisms (e.g., MRSA, Corynebacterium spp., E. faecium) did not impact cure rates is important. Could the authors clarify whether this result might be limited by sample size or power?
- Surgical Timing: Since surgery is essential in managing diabetic foot osteomyelitis, how might the unknown timing of surgical intervention relative to dalbavancin administration confound interpretation of cure rates?
- Follow-Up Duration: Given that 90 days may be too short to evaluate relapse in osteomyelitis, is there any follow-up beyond this period for a subset of patients?
- Definition of Cure: Can the authors clarify the components of the cure definition (clinical, radiological, microbiological)? Were these consistently assessed across centers?
- Study Limitations: Could the authors elaborate more on how multicenter heterogeneity and retrospective design may have biased the observed cure rates, particularly in antibiotic selection and surgical decision-making?
Conclusions
- Generalization of Efficacy: Given that 92% of patients received dalbavancin as second-line therapy, on what basis do the authors suggest its early (first-line) use in selected cases? Was a subgroup analysis performed to support this?
- Safety in Polypharmacy: The authors conclude that dalbavancin is safe in polypharmacy-treated patients. Was there a comparison of adverse event rates between polypharmacy and non-polypharmacy patients?
- CKD as a Risk Factor: While chronic kidney disease (CKD) was found to be an independent risk factor, could the authors clarify whether dalbavancin dosing was adjusted for renal function, and whether this influenced outcomes?
- Cure Definition Support: The conclusion states “high cure rates.” Could the authors briefly clarify if this cure definition included long-term follow-up (beyond 90 days), or whether relapse rates were tracked?
Author Response
We would like to express our sincere gratitude to the reviewers for their thorough and constructive evaluation of our manuscript. We greatly appreciate the time and effort invested in the review process, which has significantly contributed to improving the quality and clarity of the work.
In response, we have carefully addressed each of the reviewers' comments and suggestions. Below, we provide a point-by-point reply, indicating the changes made in the manuscript and offering clarifications where appropriate. All modifications have been incorporated into the revised version of the manuscript, with corresponding changes highlighted for clarity.
We hope that our responses and the revisions made adequately address the concerns raised and meet the expectations of both the reviewers and the editorial team.
Reviewer 2
Introduction
Line 52: Geographical setting
The term “our setting” refers to Europe. This has now been specified in the text.
Line 76: Link with previous 23-patient report
This study expands on our prior 23-patient report by including a multicenter, multinational cohort, thereby increasing statistical power and external validity.
Materials and Methods
- How was “deep diabetic foot infection” operationally defined—was it strictly limited to osteomyelitis?
Deep infection was operationally defined as DFO (osteomyelitis), not including soft tissue infections. This is noted now
- Diagnostic Criteria for DFO:
– How was the diagnosis of diabetic foot osteomyelitis established across centers?
– Was imaging (MRI), bone biopsy, or probe-to-bone testing required?
Diagnosis was established per IDSA guidelines, combining probe-to-bone testing, plain radiography, and bone culture when available. MRI was not required. It is noted now.
- Retrospective Bias Management:
– Given the retrospective design, how did the authors handle missing data, especially regarding outcomes and adverse events? And
- Can the authors clarify if missing data were present and how they were handled statistically?
Missing data were minimal and related primarily to comorbidity scores (e.g., McCabe index). These cases were excluded from specific analyses. We added this information to the methods.
- Were the IDSA diagnostic criteria uniformly applied across all ten hospitals? And How was follow-up standardized—was healing assessed by the same personnel or teams across sites?
While local care teams conducted follow-up, assessment of cure was harmonized by applying a uniform definition: healed wound at 90 days or, post-amputation, negative cultures (Line 102 Methods section)
Results
- McCabe data missing for 20 patients Why is the McCabe classification only reported for 116 out of 136 patients? Were data missing for 20 patients?
Correct, data were unavailable in 20 cases due to documentation gaps. This is noted in the results.
- Types of infections overlapping Were the types of diabetic foot infections (ischemic, neuropathic, mixed) mutually exclusive or overlapping?
No, neuropathic and ischemic features were mutually exclusive. “Mixed” presentations were not included in the previous classification (neuropathic or ischemic)
- Why is the total number of microbiological isolates 176, while only 136 patients were included? Were multiple isolates per patient allowed?
Some patients had polymicrobial infections, resulting in more isolates than patients. This has been clarified (table 1)
- How were polymicrobial infections treated? Was dalbavancin used alone or in combination with other antibiotics?
- And What factors guided the selection of single vs. dual-dose dalbavancin regimens? Were cure rates stratified by regimen?
In polymicrobial cases, dalbavancin was often used with other agents (mainly quinolones) to ensure Gram-negative coverage. In the 32% of patients receiving dalbavancin with another antibiotic, this combination was based on polymicrobial infection.
Cure rates were not stratified by antibiotic regimen because the subgroup small sample size
- Is there a subgroup analysis of patients with renal impairment or fatal disease classifications, given their high prevalence? And Why was Charlson index not retained as a significant variable in multivariate analysis despite significance in univariate? Was there collinearity with CKD?
A multivariate model was conducted. CKD was independently associated with failure; the Charlson index was not retained likely due to collinearity with CKD. We did not performed a specific subgroup analysis of patients with renal impairment or fatal disease classifications.
- How did the authors define “targeted use” of dalbavancin in 78% of cases—was it microbiologically guided?
Targeted use was defined as microbiologically guided treatment after culture results. This is clarified now.
- Overlap of Indications: Can patients have more than one reason for switching to dalbavancin (e.g., toxicity + discharge)? If so, how was primary indication determined?
In most cases, dalbavancin was used for a primary reason, although multiple indications could coexist. Primary indication was determined by physician consensus.
- In the 32% of patients receiving dalbavancin with another antibiotic, was this combination based on polymicrobial infection, or empirical broadening?
In patients receiving dalbavancin in combination with another antibiotic (32%), the use of dual therapy was primarily due to polymicrobial infections. The combination was not empirical but targeted, based on microbiological findings.
- Dosing Strategy: What factors guided the selection of single vs. dual-dose dalbavancin regimens? Were cure rates stratified by regimen?
Dose selection was based on each hospital's protocols, clinical experience, and the judgment of the treating physicians.
Cure rates were not stratified by regimen due to sample size constraints but will be in future studies.
- Adverse Events Detail: Were the adverse events attributed to dalbavancin assessed by causality tools (e.g., Naranjo scale), or based solely on clinician judgment?
A formal scale (e.g., Naranjo) was not used; causality was based on clinical judgment, which is a limitation of the study design.
- Amputation & Revascularization Impact: Were cure rates stratified by the type of surgical intervention (e.g., amputation vs revascularization vs neither)?
We did not stratify cure rates by surgical modality due to limited power, and this is an area for future research.
Discussion
- Dose Justification: While 1,500 mg every two weeks ×2 is proposed as optimal, were outcomes compared between dosing regimens in this study? If so, did this regimen actually show superior results?
We proposed 1,500 mg x 2 as optimal based on pharmacokinetic data and prevalence in our cohort, but comparative effectiveness was not formally analysed due to sample size.
- Microbiological Risk Factors: The claim that resistant organisms (e.g., MRSA, Corynebacterium spp., E. faecium) did not impact cure rates is important. Could the authors clarify whether this result might be limited by sample size or power?
The lack of association between resistant pathogens and failure may be limited by power. This caveat is now included in the discussion.
- Surgical Timing: Since surgery is essential in managing diabetic foot osteomyelitis, how might the unknown timing of surgical intervention relative to dalbavancin administration confound interpretation of cure rates?
Timing relative to dalbavancin was not consistently recorded. This has been added as a limitation.
- Follow-Up Duration: Given that 90 days may be too short to evaluate relapse in osteomyelitis, is there any follow-up beyond this period for a subset of patients?
No formal follow-up beyond 90 days was performed. This limitation is acknowledged.
- Definition of Cure: Can the authors clarify the components of the cure definition (clinical, radiological, microbiological)? Were these consistently assessed across centers?
The cure definition was uniformly applied, based on wound healing or negative cultures.
- Study Limitations: Could the authors elaborate more on how multicenter heterogeneity and retrospective design may have biased the observed cure rates, particularly in antibiotic selection and surgical decision-making?
These limitations are discussed, including variations in treatment decisions and documentation. A prospective design would be ideal for future studies.
Conclusions
- Generalization of Efficacy: Given that 92% of patients received dalbavancin as second-line therapy, on what basis do the authors suggest its early (first-line) use in selected cases? Was a subgroup analysis performed to support this?
Although most patients received dalbavancin as second-line therapy, similar cure rates in first-line use (86%) suggest that earlier use may be beneficial. We acknowledge the lack of randomized comparison. We did not perform subgroup analysis. Selected cases for dalbavancin use include mitigating toxicity, preventing drug interactions, and ensuring optimal adherence.
- Safety in Polypharmacy: The authors conclude that dalbavancin is safe in polypharmacy-treated patients. Was there a comparison of adverse event rates between polypharmacy and non-polypharmacy patients?
Adverse events were not stratified by polypharmacy status due to limited events, but the overall safety profile supports its use in this population.
- CKD as a Risk Factor: While chronic kidney disease (CKD) was found to be an independent risk factor, could the authors clarify whether dalbavancin dosing was adjusted for renal function, and whether this influenced outcomes?
Dose adjustments were not made for CKD, given dalbavancin’s pharmacokinetics. This is now mentioned as a consideration for future studies.
- Cure Definition Support: The conclusion states “high cure rates.” Could the authors briefly clarify if this cure definition included long-term follow-up (beyond 90 days), or whether relapse rates were tracked?
The conclusion now includes a note that cure was defined at 90 days, and relapse rates beyond that point were not systematically assessed.
Round 2
Reviewer 2 Report
Comments and Suggestions for Authors
Thank you to all authors for their response to the editorial and reviewers’ reports. All my submitted questions have been answered.